# The Effect of Stand Density, Biodiversity, and Spatial Structure on Stand Basal Area Increment in Natural Spruce-Fir-Broadleaf Mixed Forests

**Di Liu [1,2], Chaofan Zhou [1,2], Xiao He [1,2]** 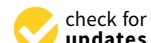, **Xiaohong Zhang [1,2], Linyan Feng [1,2] and Huiru Zhang [2,3,*]**

[1] Research Institute of Forest Resource Information Techniques, Chinese Academy of Forestry, Beijing 100091, China; liudi19920109@163.com (D.L.); cfzhou2021@163.com (C.Z.); hexiao@ifrit.ac.cn (X.H.); zhangxh@ifrit.ac.cn (X.Z.); Linyan_feng@caf.ac.cn (L.F.)

[2] Key Laboratory of Forest Management and Growth Modeling, National Forestry and Grassland Administration, Beijing 100091, China

[3] Experimental Center of Forestry in North China, Chinese Academy of Forestry, No.1 Shuizha West Road, Mentougou District, Beijing 102300, China

[*] Correspondence: huiru@caf.ac.cn; Tel.: +86-10-69836348

**Abstract:** Forest trees exhibit a large variation in the basal area increment (BAI), and the variation is attributed to the stand density, biodiversity, and stand spatial structure. Studying and quantifying the effect of these above variables on tree growth is vital for future forest management. However, the stand spatial structure based on neighboring trees has rarely been considered, especially in the mixed forests. This study adopted the random-forest (RF) algorithm to model and interpret BAI based on stand density, biodiversity, and spatial structure. Fourteen independent variables, including two stand density predictors, four biodiversity predictors, and eight spatial structure predictors, were evaluated. The RF model was trained for the whole stand, three tree species groups (gap, neutral, and shade_tolerant), and two tree species (spruce and fir). A 10-fold blocked cross-validation was then used to optimize the hyper-parameters and evaluate the models. The squared correlation coefficients ($R^2$) for the six groups were 0.233 for the whole stand, 0.575 for fir, 0.609 for shade_tolerant, 0.622 for neutral, 0.722 for gap, and 0.730 for spruce. The Stand Density Index (*SDI*) was the most-important predictor, suggesting that BAI is primarily restricted by competition. BAI and species biodiversity were positively correlated for the whole stand. The stands were expected to be randomly distributed based on the relationship between the uniform angle index (W) and growth. The relationship between dominance (U) and BAI indicated that small trees should be planted around the light-demanding tree species and vice versa. Of note, these findings emphasize the need to consider the three types of variables in mixed forests, especially the spatial structure factors. This study may help make significant advances in species composition, spatial arrangement, and the sustainable development of mixed forests.

**Keywords:** spatial structure parameters; stand density; biodiversity; random forests; BAI model

## 1. Introduction

Understanding forest growth is crucial for forest management and the estimations of net primary production and carbon sequestration [1]. Forests with mixed species have higher productivity, higher temporal stability, a lower risk of biotic and abiotic disturbances, and a more diverse ecosystem. However, it is challenging to model their growth because of complex communities [2]. Basal area growth increment (BAI) is particularly suitable for modelling tree growth among the other measurements because it is directly related to the diameter at breast height (DBH), thus making it more reliable [3,4]. Machine-learning algorithms have already been used in forest management, providing more accurate predictions than the traditional linear regressions in dendroclimatology [5–9]. Notably, Martin Jung used BAI modelling of spruce and fir in random forests [10].

The role of stand biodiversity for ecosystem processes, such as biomass production, nutrient storage as well as energy and material flow, is currently the most demanding ecological research topic, especially in the context of climate change [11]. Numerous indicators have been developed to measure species and structural diversity to enable local and temporal comparisons [12]. Studies of the relationship between forest diversity and productivity in various forest ecosystems have revealed positive, neutral, and negative correlations [13]. Moreover, the impact of species diversity on productivity is controversial, especially in forest communities with complicated spatial structures and long life spans of dominant organisms [14]. The scale and stand development stage evaluated must therefore be considered when discussing diversity [15,16]. Density is also a foundational component of forest growth models. Density measurements combining the use of growing areas with the size of the trees are required when comparing stands of different forest ages or different locations [17]. Strikingly, the Stand Density Index (SDI) proposed by Reineke (1933) is the most commonly used measurement method. It accounts for the number and average diameter of trees based on the allometric relationship between these two variables to express the tree number as a reference quadratic mean diameter [18]. The SDI has been used in a great deal of forest growth models for its simplicity [19–21]. However, there are very limited studies available regarding SDI on BAI, and additional research is needed.

The forest structure is a dominant feature of forests, reflecting the connection among individual trees, their regeneration patterns, and their competition. It thus plays a crucial role in the natural dynamics of ecosystem [22–24]. There are various methods available for measuring forest structure. Among them, a quantization method based on a basic unit (a reference tree with its four nearest trees) is accepted and widely used in quantifying forest structure [25–27], which is defined as spatial structural unit. The uniform angle index (W) [28], dominance (U) [29], mingling (M) [30], and crowding (C) [31] can be calculated based on the spatial structural unit. The four structural parameters accurately express the relative size of individuals, the spatial pattern of neighbor trees, tree species mingling, and whether they have horizontal competition [32]. The optimal stand spatial state is a significant factor to the development of forest growth [33]. Some cross-sectional studies found that individual density of neighboring trees or the same species, the size distribution, and the number of relative species significantly associate with the growth of target trees [34,35]. However, there have been surprisingly few historical studies on the relationship between growth and spatial structure through the model.

This study aimed to (1) explore and evaluate a machine-learning model for BAI predictions for the whole stand, tree species. and tree species groups in the north-eastern area of China; (2) identify the main predictor variables; and (3) interpret the effect of key predictors of forest structure on BAI and provide guidance for forest management.

## 2. Materials and Methods

### 2.1. Study Area and Experimental Design

The study area was located in the Jingouling Forest Farm in the northeast of Wangqing County, Jilin Province, in the low hilly area of the Changbai Mountains. The altitude of the study area is 300–1200 m, with its slope is mainly concentrated at 5°–25°. The region has a temperate continental monsoon climate with four distinct seasons, with temperatures ranging from −32 °C to 32 °C and annual average rainfall from 600 mm to 700 mm. The early frost starts in mid-September, and the late frost extends to the end of May of the following year. The growth period is 120 days, and the average snow cover is 50 cm thick. The coniferous gray-brown loamy soil is dominant, with a thickness of about 45 cm harboring high natural fertility. The parent rock is basalt, with a moist and loose granular structure and an acidic pH. This study adopted spruce-fir natural forest as an example, with the study area being a typical representative of over-deforestation.

One-hectare sampling plots (100 m × 100 m) were established in a spruce-fir-broadleaf mixed forest in 2013 (Figure 1 and Table 1). All live trees with diameters at breast height (DBH) ≥ 1 cm in each plot were recorded. Their locations were then mapped, their species

identified, and their DBH, height (H), crown diameter, and clear bole height were measured. Remeasurements were done in 2018, five years after the study began. Three control plots were then selected and divided into 75 small plots, measuring 20 × 20 m. The translation method is used for edge-correction [36]. Tree samples with ≥5 cm diameter at breast height (DBH) were then used to calculate the stand basal area, spatial-structure parameters, species-diversity indices, and stand-structure indices. The remaining records were finally obtained for analysis, excluding the plots where the stand basal area was inverted. Tree data collected from 2013 to 2018 were used for stand basal area prediction.

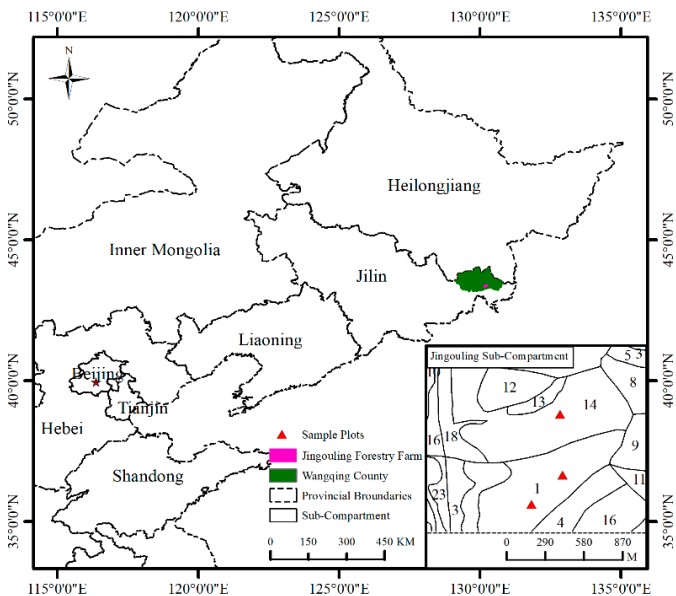

**Figure 1.** Location of study area: Wangqing Forest Bureau in northeast China and spatial distribution of 3 sample plots (bottom).

**Table 1.** Stand characteristic of 3 plots.

| Plot Code | Stem Number /ha | Mean DBH/ cm | Basal Area/ ($m^2$/ha) | Stock Volume/ ($m^3$/ha) | Canopy Density | Ba Growth in 5 Years/ ($m^2$/ha) |
|---|---|---|---|---|---|---|
| 1 | 996 | 17.63 | 24.30 | 199.97 | 0.85 | 1.85 |
| 2 | 1024 | 18.23 | 26.72 | 216.00 | 0.86 | 2.12 |
| 3 | 1018 | 17.38 | 24.15 | 182.75 | 0.63 | 2.55 |

### 2.2. Predictor Variables

The predictor variables were consolidated into three groups: (1) spatial-structure parameters, (2) biodiversity indices, and (3) forest stand characteristics. Four forest spatial-structure parameters based on the spatial structural unit were used as the spatial-structure parameters. They included the uniform angle index (W), describing the degree of tree distribution uniformity; dominance (U), reflecting the degree of DBH differentiation; mingling (M), expressing the degree of tree species segregation; and crowding (C), reflecting the degree of tree crowding. Tree species diversity, tree size diversity, and the integrated diversity index of tree species and size were used as the biodiversity indices, whilst Reineke's stand density index (SDI) was used to measure the stand characteristics. Table 2 highlights the final set of independent variables with descriptive statistics.

**Table 2.** List of independent variables and description.

| | Variable Name | Formula | Description |
|---|---|---|---|
| Biodiversity | Tree species diversity | $Div\_SP = -\sum \dfrac{N_i}{N}\left(\ln\dfrac{N_i}{N}\right)$ | $N_i$ is the number of trees in the *i*-th tree species; $N$ is the total number of trees. |
| | Tree size diversity | $Div\_Size = -\sum \dfrac{N_j}{N}\left(\ln\dfrac{N_j}{N}\right)$ | $N_j$ is the number of trees in the *j*-th diameter class; $N$ is the same as above. When $N_j$ and $N$ are the number of trees in *j*-th diameter class and the total of a tree species group or a tree species, respectively, the index is going to be *Div_Size_Part*. |
| | Integrated diversity index of tree species and size | $Div\_SPSize = -\sum \dfrac{N_{ij}}{N}\left(\ln\dfrac{N_{ij}}{N}\right)$ | $N_{ij}$ is the number of trees of the *j*-th diameter class in the *i*-th tree species; $N$ is same as above. |
| Stand density | Reineke's stand density index | $SDI = N\left(\dfrac{D}{D_0}\right)^b$ | $N$ is same as above; $D$ is the actual average diameter of the stand; $D_0$ is the standard average diameter of 20 cm; $b$ is the natural thinning slope coefficient of 1.605. When $D$ and $N$ are the actual average diameter and the number of trees of a tree species group or a tree species, respectively, the index is going to be *SDI_Part*. |
| Spatial-structure indices | Uniform angle index | $Wm = \dfrac{1}{M}\sum_{i=1}^{M}\dfrac{1}{4}\sum_{j=1}^{4} z_{ij}$ <br> $z_{ij} = \begin{cases} 1, & \text{if } \alpha_j < \alpha_0 \\ 0, & \text{otherwise} \end{cases}$ | $M$ is the number of target trees; *i* is the *i*-th target tree; *j* is the *j*-th adjacent tree of the *i*-th target tree; $z_{ij}$ is the judgment result of angles with standard angle; $\alpha_j$ is the *j*-th angle formed by the *j*-th adjacent tree and the previous one; $\alpha_0$ is the standard angle of 72°. When $M$ is the number of target trees of a tree species group or a tree species, the index is going to be *Wm_Part*. |
| | Dominance | $Um = \dfrac{1}{M}\sum_{i=1}^{M}\dfrac{1}{4}\sum_{j=1}^{4} k_{ij}$ <br> $k_{ij} = \begin{cases} 1, & \text{if } DBH_j \geq DBH_i \\ 0, & \text{otherwise} \end{cases}$ | $M$, *i*, and *j* are same as above; $k_{ij}$ represents the judgment result of diameters; $DBH_j$ is the DBH of the *j*-th adjacent tree; $DBH_i$ is the DBH of the *i*-th object tree. When $M$ is the number of target trees of a tree species group or a tree species, the index is going to be *Um_Part*. |
| | Mingling | $Mm = \dfrac{1}{M}\sum_{i=1}^{M}\dfrac{1}{4}\sum_{j=1}^{4} v_{ij}$ <br> $v_{ij} = \begin{cases} 1, & \text{if } sp_j \neq sp_i \\ 0, & \text{otherwise} \end{cases}$ | $M$, *i*, and *j* are same as above; $v_{ij}$ is the judgment result of tree species codes; $sp_j$ is the tree species code of the *j*-th adjacent tree; $sp_i$ is the tree species code of the *i*-th object tree. When $M$ is the number of target trees of a tree species group or a tree species, the index is going to be *Mm_Part*. |
| | Crowding | $Cm = \dfrac{1}{M}\sum_{i=1}^{M}\dfrac{1}{4}\sum_{j=1}^{4} y_{ij}$ <br> $y_{ij} = \begin{cases} 1, & \text{if } c_j + c_i > dist_{ij} \\ 0, & \text{otherwise} \end{cases}$ | $M$, *i*, and *j* are same as above; $y_{ij}$ is the judgment result of two crown radii with distance; $c_j$ is the crown radius of the *j*-th adjacent tree; $c_i$ is the crown radius of the *i*-th object tree; $dist_{ij}$ is the distance between the *j*-th adjacent tree and the *i*-th object tree. When $M$ is the number of target trees of a tree species group or a tree species, the index is going to be *Cm_Part*. |

*2.3. Random-Forest Algorithm for Predicting Stand Basal Area Increment (BAI)*

The random forest (RF), a nonparametric, data-driven method based on an ensemble of bootstrapped classification or regression trees, was the primary modelling tool used in this study. The excellent predictive power of the random-forest algorithm is based on the construction of a random regression tree for each bootstrapping sample. At each tree node, only a subset of all nodes is divided, taking into account the available independent variables. This subset is defined with the parameter mtry. The mtry specifies the number of variables to randomly selected and tested at each node of each tree. The tool employs three steps to implement the regression: (1) Bootstrap is used to sample the original training set (n × p) to get k training sets (n × p); (2) a regression tree model for k training sets is then built separately to get k regression results; (3) finally, the average of k regression results is taken as the final prediction result [37]. RF improves prediction accuracy by integrating multiple regression trees; it has the ability to fit nonlinear functions; and it can improve the interpretability of the model by measuring the relative importance of independent variables [38]. We used the R RandomForest package to implement the RF algorithm for modelling BAI [39].

Predictive BAI models were developed for six groups: whole stand, gap, neutral, shade_tolerant, spruce (*Picea jezoensis*), and fir (*Abies nephrolepis*), with gap stands for light-demanding tree species, neutral stands for neutral tree species, and shade_tolerant stands for shade-tolerant tree species (Table 3).

**Table 3.** List of tree species group classification.

| Species Group | Tree Species |
|---|---|
| Gap | Asian white birch (*Betula platyphylla*), Korean pine (*Pinus koraiensis*), Changbai larch (*Larix olgensis*), Ussuri popular (*Populus ussuriensis*), elm (*Ulmus japonica).* |
| Neutral | Linden (*Tilia amurensis*), ribbed birch (*Betula costata*), maple (*Acer mono*), ash (*Fraxinus mandschurica*). |
| Shade_tolerant | Corktree (*Phellodendron amurense*), fir (*Abies nephrolepis*), spruce (*Picea jezoensis*). |

The forest stand density, biodiversity, and spatial-structure factors were used as input variables of RF to establish the BAI regression model. The variance inflation factor was first used to determine the collinearity and to remove the variables from the final model accordingly (variance inflation factor >10) to exclude strongly related variables and to avoid introducing unnecessary parameters [40]. All predictor variables were used to train the RF model for each group independently based on the value of mtry parameters between 2 and 14. A 10-fold blocked cross-validation was then used to optimize hyper-parameters and to evaluate the BAI models. The squared correlation coefficient ($R^2$) and the mean square error (RMSE) were averaged across the 10 validation folds. For each group, the highest $R^2$ was selected and used for further analysis.

## 3. Results

*3.1. Random Forest Model Evaluation*

Spruce had the highest performance ($R^2$: 0.730), while the whole stand had the lowest performance ($R^2$: 0.223) (Table 4). In general, the tree species group (gap with mtry = 8, neutral with mtry = 10, and shade_tolerant with mtry = 14) and the tree species (spruce with mtry = 11 and fir with mtry = 14) had higher fitting accuracy than the whole stand.

**Table 4.** Ten-fold blocked cross-validation results for the six groups.

| Groups | mtry | $R^2 \pm$ std | RMSE $\pm$ std (m$^2$/ha) |
|---|---|---|---|
| Whole stand | 8 | $0.223 \pm 0.468$ | $0.534 \pm 0.132$ |
| Gap | 8 | $0.722 \pm 0.207$ | $0.236 \pm 0.063$ |
| Neutral | 10 | $0.622 \pm 0.306$ | $0.231 \pm 0.066$ |
| Shade_tolerant | 14 | $0.609 \pm 0.295$ | $0.167 \pm 0.074$ |
| Spruce | 11 | $0.730 \pm 0.214$ | $0.061 \pm 0.038$ |
| Fir | 14 | $0.575 \pm 0.282$ | $0.157 \pm 0.070$ |

Footer: mtry stands for optimized random forest parameter; $R^2$ denotes the squared correlation coefficient; RMSE denotes the root mean square error; and std stands for the standard deviation.

### 3.2. The Relative Importance (%) of Predictors

The most-important predictors of each variable were determined based on performance during prediction (Table 5). *SDI* ranked as the top discriminative predictor for the whole stand, demonstrating its effectiveness for BAI. Four spatial-structure predictors explained 51.86% of all in total. *Um* had the largest proportion (21.18%), explaining almost the same amount of *SDI*. *SDI* explained 50.69%, while biodiversity explained 22.41% of all in the gap. As regards spatial-structure predictors, *Mm_part* had the largest proportion (7.97%) among the spatial-structure predictors. In the same line, *SDI* explained 38.28%, 53.27%, and 52.58% for neutral, shade_tolerant, and spruce, respectively. Compared with gap, *Um_part* (8.56%) accounted for the largest proportion ahead of *Mm_part* (8.04%) in shade_tolerant. However, *SDI* (24.98%) had the largest contribution than *Mm_part* (15.37%) in fir.

**Table 5.** The relative importance (%) of predictors for the six groups.

| Predictors | Whole Stand | Gap | Neutral | Shade_Tolerant | Spruce | Fir |
|---|---|---|---|---|---|---|
| *Wm* | 11.61 | 2.27 | 3.01 | 2.29 | 2.19 | 4.74 |
| *Um* | 21.18 | 2.91 | 3.33 | 4.61 | 2.21 | 2.90 |
| *Mm* | 8.76 | 3.96 | 4.17 | 3.10 | 3.80 | 8.36 |
| *Cm* | 10.30 | / | / | 1.62 | 1.12 | 3.44 |
| *SDI* | 21.83 | 2.45 | 5.74 | 2.44 | 1.53 | 8.43 |
| *Div_SP* | 6.66 | 7.51 | 3.65 | 2.33 | 3.08 | 6.21 |
| *Div_SPSize* | 10.63 | 4.76 | 5.84 | 1.95 | / | 3.27 |
| *Div_Size* | 9.02 | 2.17 | 3.60 | 2.92 | 2.86 | 1.84 |
| *SDI_part* | / | 50.69 | 38.28 | 53.27 | 52.58 | 24.98 |
| *Div_Size_part* | / | 7.97 | 7.83 | 3.07 | 11.50 | 6.34 |
| *Wm_part* | / | 3.02 | 5.80 | 2.85 | 0.94 | 1.34 |
| *Um_part* | / | 2.40 | 5.29 | 8.56 | 4.49 | 8.18 |
| *Mm_part* | / | 6.84 | 5.03 | 8.04 | 12.73 | 15.37 |
| *Cm_part* | / | 3.07 | 8.42 | 2.97 | 0.97 | 4.61 |

### 3.3. The Effects of Predictors on BAI

*SDI* was the most-important variable for BAI for whole stand, gap, neutral, shade_tolerant, fir, and spruce. It had a strong positive effect on BAI, but when *SDI* in whole stand (Figure 2) and *SDI_part* in tree species groups (Figure 3) or tree species (Figure 4) reached the threshold, BAI stopped increasing and became a constant. For whole stand, gap, neutral, and shade_tolerant, the inflection points were approximately 1000, 500, 500, and 300, respectively. For fir and spruce, this point was reached relatively early (Figures 2–4).

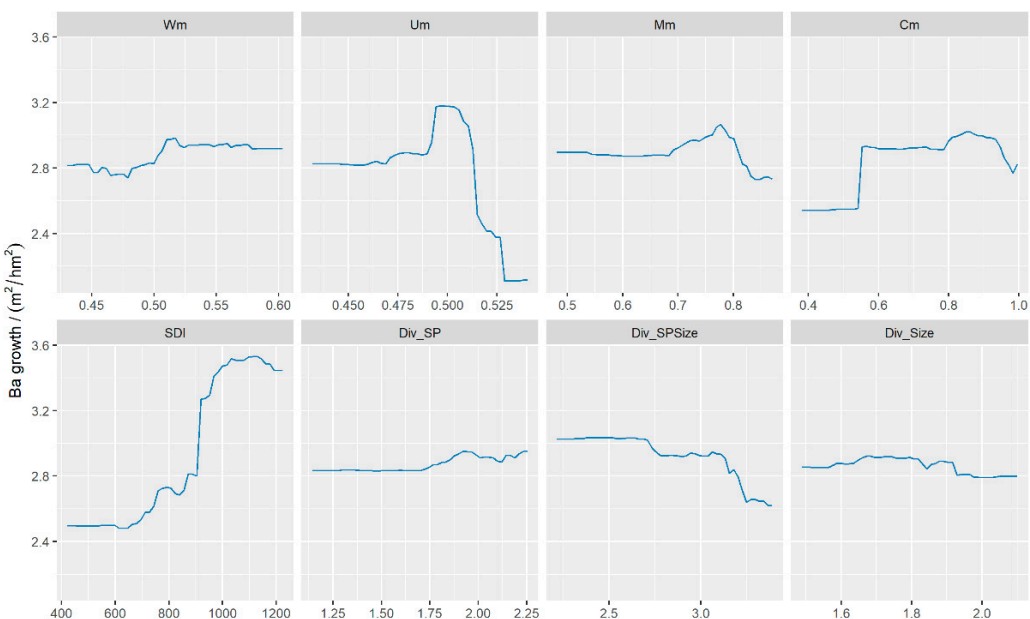

**Figure 2.** Basal area increment (BAI) of whole-stand simulations in relation to the most important predictor variables.

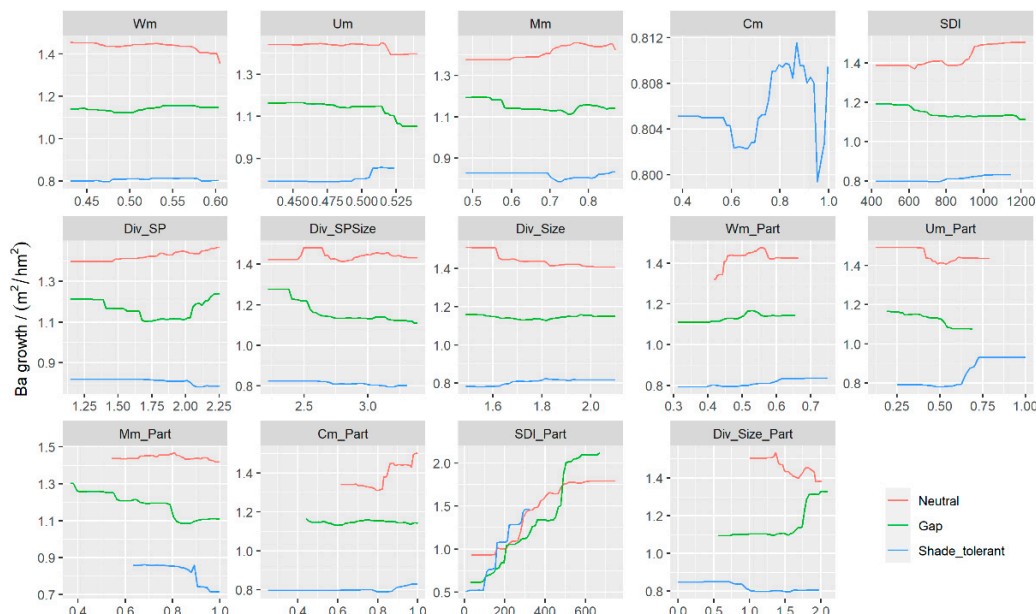

**Figure 3.** Basal area increment (BAI) of tree-species-groups simulations in relation to the most-important predictor variables.

For whole stand (Figure 2), *Um* was the most-important predictor of the stand spatial structure. BAI first increased and surpassed a maximum and then decreased as *Um* increased. BAI increased with the increase in *Mm* and surpassed a maximum and then decreased. BAI increased slowly with the increase in *Wm* and then became constant at a maximum value of 0.517. However, BAI increased sharply when *Cm* took a value between 0.5 and 0.6. When it came to *Div_SPSize*, there was a clear trend of decreasing at the late of the curve. In contrast, *Div_Size* had little discernible influence on BAI. When *Div_SP* was between 1.75 to 2.25, BAI increased gradually with a slight fluctuation.

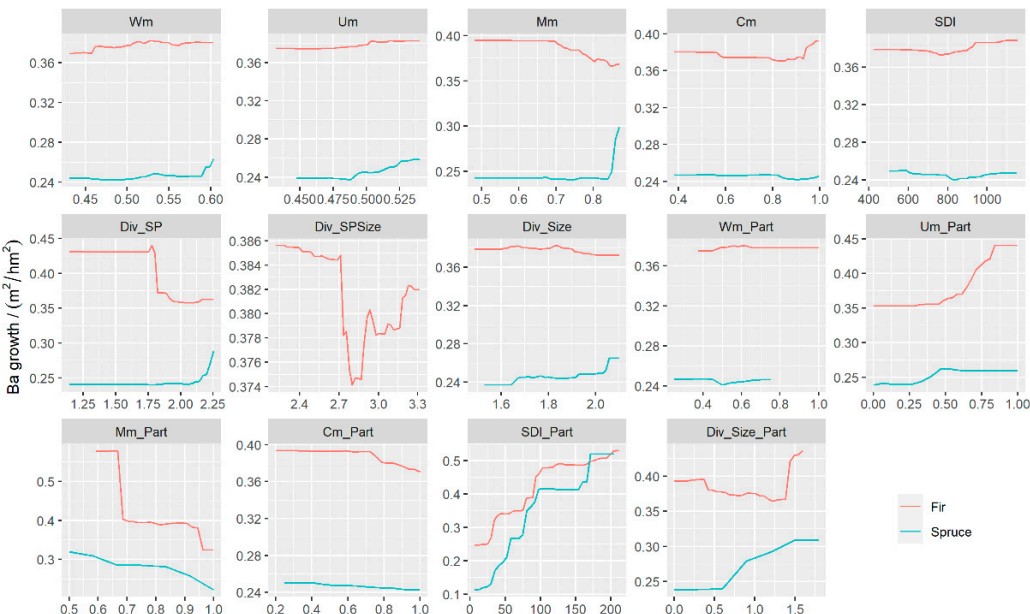

**Figure 4.** Basal area increment (BAI) of tree species simulations in relation to the most-important predictor variables.

When considering *Wm_part*, neutral had a greater impact on BAI than gap and shade_tolerant. The BAI of gap decreased first and then gradually stabilized with the increase in *Um_Part* in gap, while the BAI of shade_tolerant increased first (after a period of quiescent stability) and then tended to be stable with the increase in *Um_Part* in shade_tolerant (Figure 3). *Cm_Part* of neutral with values predominantly distributed among 0.6 and 1.0, which was positively correlated with the BAI of neutral. By comparison, *Cm_Part* of gap and *Cm_Part* of shade_tolerant had less influence on BAI. When it came to *Mm_Part*, the relationship between BAI and neutral was relatively weak; however, BAI was negatively correlated with gap and shade_tolerant. The effects of the four diversity indicators on shade_tolerant were negligible. For *Div_Size_Part*, the effects of gap and neutral were inconsistent.

The BAI of fir and spruce first stayed constant and then increased and finally stabilized with the increase in *Um_part* (Figure 4). Although fir and spruce of *Mm_Part* were negatively correlated with BAI, the influence of fir was more significant. The BAI of fir did not change with *Div_SP* at the beginning, and when the value of *Div_SP* reached 1.75, BAI dropped. However, the effect of *Div_SP* on the BAI of spruce could be negligible. The response of fir BAI and spruce BAI to *Div_Size_Part* were inconsistent. Consistent with *Wm_part*, *Cm_part* also promoted a very weak response to the BAI in fir and spruce.

## 4. Discussion

### 4.1. Evaluation of Random-Forest Model

The explained variance is usually lower in mixed-species unevenly aged forests than that in even-aged pure stands [2]. This result is attributed to the significant differences in the growth of various tree species in the mixed forest, which causes difficulties in prediction. In our study, tree species (spruce and fir) and tree species group (gap, neutral, and shade_tolerant) had better results than the whole stand in BAI modelling. This finding was attributed to the varying responses of different tree species to spatial structure, stand structure, and density [41]. A better simulation effect was further achieved by dividing the forest stand into tree-species groups. This view was supported by Ni et al. [42], who also found that the division is necessary for analyzing the mixed species stand by grouping species.



*4.2. The Effects of STAND Density and Biodiversity on BAI*

Stand BAI tends to increase with an increase in stand density and then stabilizes after the tree crown has peaked [43]. *SDI* (Figure 2) or *SDI_part* (Figures 3 and 4) follows the same trend in the nature mixed forests. The reason for this and the possible underlying mechanism were investigated. Tree growth is rarely restricted by competition during early stand development because the resources are sufficient. Increasing dominance near canopy closure reduces the efficiency of resource use by non-dominant trees, lowering overall stand growth. The hierarchical structure of the tree during late stand development creates a spatial heterogeneity of the light resources, causing several species to coexist by compromising their demographic parameters.

Tree species diversity, tree size diversity, and the integrated diversity index of tree species and size were used in our study. The positive correlation observed between the BAI and species diversity (Figure 2) was consistent with the findings of several previous studies [44]. This effect was related to full utilization of natural resources in stands with higher tree species diversity. The trends of tree species diversity and integrated diversity index of tree species and size were opposite at the whole stand level (Figure 2). Moreover, the relationship between the BAI and the integrated diversity index of tree species and size was less clear, whereas the BAI was weakly correlated with tree size diversity (Figure 2), essentially because the largest trees do not counterbalance the lesser growth of the smaller trees in uneven-aged forests compared with even-aged forests.

*4.3. Simulated Effect of Forest Spatial Structure Variables on BAI Prediction*

The growth and production of mixed forest stands have become a popular research topic as these types of forests have functional and service advantages [17]. Spatial structures are of paramount importance in community research because their existence suggests their formation process [45]. The importance of structure on forest processes highlights the value of understanding the relationship between stand structure and growth and how this relationship is influenced by tree crowding and relative tree size, species mixing, and spatial distribution. Analyzing the growth of mixed stands requires a modelling approach that relates the growth of individual trees to their resource availability and provides output values per species [46]. In recent years, a group of neighborhood-based structural parameters with strong functionality that can fully express the spatial-structural features of tree populations and forest communities, and that can guide forestry practices, has attracted widespread attention [47]. Previous studies have mainly focused on the univariate distribution and mean value of the structural parameters to analyze the structural properties of stands. Several studies have further analyzed the influence of a horizontal spatial distribution and species mingling on stand growth in mixed forest using individual tree models [48,49]. However, only a few empirical-based studies have been conducted to assess these effects at the stand level. Neighboring trees may be able to improve the predictions of the model. Models with spatially explicit structural indices perform better than those with non-spatial indices due to the complexity of the stand structure [50]. Furthermore, relationships between stand growth and a given structural variable can vary widely depending on the forest types and species [38].

Spatial-distribution patterns reflect the way individuals gather or disperse on the horizon and determine the distribution of microclimatic conditions, the availability of resources, and the formation of habitat niches; thus, they significantly influence the biological diversity, tree growth, and timber production [51]. Generally, the spatial structures of natural forests always follow random distributions in horizontal space. Such a spatial distribution may be quantified using an area-independent nearest-neighborhood uniform-angle index (W) proposed by Gadow and Hui (1999), and if the W-mean belongs to the confidence interval (0.475, 0.517), the distribution pattern is random; otherwise, the pattern is classified as clumped (W-mean > 0.517) or regular (W-mean < 0.475) [25]. Our results suggest that the maximum value of *Wm* is 0.517 at the whole stand level, and this value is in the range of a random distribution (0.475, 0.517). It confirms that a random distribution

is beneficial to the accumulation of forest stand area. Additionally, Zhang (2018) showed that random-structure units provide the majority of the BAI, whereas uniform and cluster structure units account only for a small part of the basal area [21].

Dominance (U) between trees plays a significant role within a stand and affects the fate of each plant. The spatial heterogeneity of light resources created by a tree hierarchy allows multiple species to coexist by means of trade-offs between demographic parameters [52]. Stand growth may increase, decrease, or remain unchanged with U, which has the advantage of quantifying plant growth among trees of different sizes within the stand [38]. U can be simply described as the number of large trees around the subject tree. One interesting finding is the BAI of gap decreases first and then gradually stabilizes at 0.5 with the increases in *Um_Part* in gap, and the BAI of shade_tolerant increases first and then tends to be stable at 0.75 with the increases in *Um_Part* in shade_tolerant. This means that if a light-demanding tree species is used as a target tree, its BAI becomes lager when its neighboring tree is a small tree. However, the shade-tolerant tree species expects the number of large trees around it to be greater than that of small trees. The growth of the light-demanding tree species needs light, so the small trees surrounding it that do not block it are more suitable for its growth, while the surrounding large trees can provide a canopy environment for shade-tolerant tree species, which is more conducive to its growth. Our study proves that the light-demanding tree species has a large BAI in the open, and the shade-tolerant tree species has a large BAI when the canopy is covered from the perspective of spatial-structure parameters.

Species mingling refers to the segregation of different species in the same community, which can describe the dissimilarity of tree species based on a reference tree and a set of neighboring trees, and the degree of mixing determines the state of light and the composition of litter, which in turn influences the growth and regeneration of trees [53]. The species-mingling value at the tree level varies inversely with the relative density of each tree species. The more uniform the trees are collected at the stand level, the lower the mixing rate between species. The species-mingling index has been used to determine the influence of structure-based forest-management methods on the spatial structure of forest stands. The focus has been put on the effects of species mixing on growth and the interactions between trees that cause such effects [13]. However, the effects of species mixing on growth largely depend on stand structure. In this contribution, species-mixing and BAI were negatively correlated in all cases except whole stand, suggesting trees with similar properties occurring in groups rather than alone have a larger BAI. However, for the whole stand, with the increases in *Mm*, BAI first increases, passes a maximum, and then decreases. This result clearly shows that it is not that the higher the degree of mixing, the greater the growth of the forest, but that there is a threshold, that is, the maximum growth of the forest is when there are three different trees around the target tree. Another important point here is that the implication of mingling (M) based on neighboring trees and species diversity are distinguished.

The degree of stand crowdedness is easy and convenient to evaluate and determines whether the forest requires to be cultivated, the time for the first thinning, and the cutting intensity. The crowding degree (C) of trees based on the relationship of neighboring trees was proposed by Hu and Hui [28]. Only limited information was found in the literature on C. Notably, the neutral tree species has the greatest correlation between BAI and *Cm*. This finding may mean that neutral tree species are the most sensitive to competition. Nevertheless, further work is required to establish the relationship between growth and crowding.

Collectively, (1) BAI and species diversity are positively correlated at the stand level, and (2) BAI first increases and then stabilizes with the increases in *SDI*. (3) The maximum value of *Wm* is 0.517. (4) BAI first increases, surpasses the maximum, and then decreases with the increases in *Mm*. At the tree level, the BAI of the gap decreases first and then gradually stabilizes with an increase of *Um* in gap, and the BAI of shade_tolerant increases first (after a period of quiescent stability) and then tend to be stable with the increases in

*Um* in shade_tolerant. Thus, our study supplies guidelines for forest management. For example, according to the relationship between biodiversity and growth, at the level of the forest stand, we need to maintain multi-species mixing. According to the relationship between *SDI* and growth, stand competition will reduce stand growth after the canopy is closed. According to the relationship between the degree of mixing and growth, the best choice for the target tree is to be adjacent to three trees of different species. According to the relationship between the uniform angle index and growth, the stands are expected to be randomly distributed. According to the relationship between dominance and growth, for light-demanding tree species, we plant small trees around it, and the opposite is true for shade-tolerant tree species. In particular, priority should be given to indicators with high importance when there are contradictions between different indicators. The simulation effect of this study may not be particularly good because it did not consider BAL (the basal area in larger trees) and other related variables, but it mainly focused on the relationship between growth and spatial structure.

## 5. Conclusions

This study presented a random-forest model for BAI estimation, which can effectively identify the relationship between stand density, biodiversity, and spatial structure on growth. The model can help in determining the most-important predictors and how they affect BAI: (1) *SDI* was the most-important predictor in BAI model positively correlated with growth. (2) A positive correlation was found between BAI and tree species diversity at the stand level. (3) The stands were expected to be randomly distributed based on the relationship between the uniform angle index (W) and growth. (4) The relationship between U and BAI indicated that small trees should be planted around the light-demanding tree species and vice versa. Undoubtedly, structure-based approaches to forest management can possibly increase forest production through changes in tree relative size and spatial-distribution patterns. Simultaneously, our results provide insight on the effects of biodiversity and density on BAI, which can assist future forest management.

**Author Contributions:** Methodology, X.H. and C.Z.; formal analysis, C.Z. and D.L.; data curation C.Z. and D.L.; writing—original draft preparation, D.L.; writing—review and editing, C.Z., X.Z. and H.Z.; visualization, C.Z. and L.F.; software, C.Z., X.H. and D.L., resources H.Z.; supervision, H.Z. All authors have read and agreed to the published version of the manuscript.

**Funding:** This research was funded by Thirteenth Five-year Plan Pioneering project of the High Technology Plan of the National Department of Technology (No.2017YFC0504101).

**Conflicts of Interest:** The authors declare no conflict of interest.

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
