# Peer review of "The Effect of Stand Density, Biodiversity, and Spatial Structure on Stand Basal Area Increment in Natural Spruce-Fir-Broadleaf Mixed Forests"

_forests, doi:10.3390/f13020162_

Round 1

Reviewer 1 Report

The paper is interesting, but it needs to be improved further:

Abstract-If possible try to sinthetize the abstract.

Introduction

L47-61-Please add more references at international level.

L78-81-Please highlight and better desribe the main contributions of this study.

Methods-Well designed and described.

Conclusions-Need to be improved since they are very resumed.

Reviewer 2 Report

The study was conducted with sufficient data and there is no critical flaw in the method. However, it seems that the description of the result, as well as the discussion, is not appropriate. The followings are the major comments:

  1. In the result section, the description of Figures 2, 3, and 4 (lines 179-186, 190-198, and 202-207, respectively) is not appropriate.  That is, the authors only commented on the results of the parameters that are convenient for the authors to support the assumption although there is no sufficient examination on the parameters those not support the authors' assumption with such comments that "those were inconsistent".
  2. In the discussion section, I think that all of the deductions should be accompanied by correspondent parts of the figures or the tables.   For example, in the "4.2 The effect of stand density and biodiversity on BAI" section (lines 224-240), it is unclear that which remarks were based on the present study's or not.
  3. In the "4.3 Simulated effect of forest spatial structure variables on BAI prediction" section, the authors insisted that "Our results suggest that the maximum value of Wm is 0.517 at the whole stand level and this value is in the range 270 of random distribution [0.475, 0.517]." (lines 269-271).  Such detailed figures, i.e., 0.517 and 0.475, would not have general meaning.  This comment also could be applied to the sentence "But for whole stand, with the increases of Mm, BAI first increases, passes a maximum around 0.775 and then decrease." in lines 305-307.

In addition, there are two minor comments:

  1. In the abstract, the abbreviation "SDI" should be explained in full words.
  2. In Figure 1, a scale should be added also for the inset figure at the bottom right.
  3. An appropriate explanation for the word "mtry parameters" in line 143 should be added combined with an additional summary on the random forest method.

That is all and thank you for your contribution.

Round 2

Reviewer 2 Report

Thank you for the revising work.  I have confirmed that all the commented points were fairly reflected over the revised manuscript.  However, the reference number larger than 18 should be carefully checked and collected because most of them in the main body are not changed although the references 19, 20 and 21 have been newly added.